# *TFM-Retouche*: A Lightweight Input-Space Adapter for Tabular Foundation Models

## Abstract

Tabular foundation models (TFMs) such as TabPFN-2.6, TabICLv2, ConTextTab, Mitra, LimiX, and TabDPT achieve strong zero-shot performance through in-context learning, but their inductive biases remain fixed at inference time. Adapting a pretrained TFM to a specific dataset typically requires either full fine-tuning, which is expensive, or parameter-efficient methods such as LoRA, which must be tailored to the internal architecture of each TFM, with mixed empirical evidence on accuracy and calibration (Tanna et al., 2026; Rubachev et al., 2025). We introduce *TFM-Retouche*, a lightweight *input-space* residual adapter that is architecture-agnostic with respect to the frozen TFM backbone. The adapter learns a small residual correction in the input space to align the input data with the inductive biases of the pretrained model, and is trained end-to-end through the frozen TFM with a post-training identity guard that falls back to the unmodified TFM whenever adaptation does not help on held-out validation. On TabArena-Lite (Erickson et al., 2025) (51 datasets spanning binary classification, multiclass, and regression), *TabICLv2-Retouche*, the framework instantiated on TabICLv2, is the top-ranked method on the leaderboard with light per-task tuning and ensembling, lifting aggregate Elo by $+56$ over the frozen TabICLv2 base and sitting on the Pareto front of predictive quality versus both training and inference time.

## 1. Introduction

Tabular foundation models (TFMs) such as TabPFN (Hollmann et al., 2023), TabPFNv2 (Hollmann et al., 2025), TabPFN-2.5/2.6 (Grinsztajn et al., 2026; 2025), TabICL/TabICLv2 (Qu et al., 2025; 2026), Mitra (Zhang et al., 2025a), LimiX (Zhang et al., 2025b), ConText-Tab (Spinaci et al., 2025), and TabDPT (Ma et al., 2026) have reshaped tabular prediction by encoding a transferable inductive bias for tabular tasks at pretraining time and predicting zero-shot through in-context learning (ICL): a labeled context is appended to the query and the prediction emerges from a single frozen forward pass (Breugel & Schaar, 2024). They often rival or surpass tuned tree ensembles on small-to-medium problems (Erickson et al., 2025), but a residual gap remains: a TFM's pretrained prior is built from generic synthetic data and is not specialized to any particular downstream task. Zero-shot ICL therefore misses potential accuracy and calibration gains that per-task adaptation can deliver.

Existing approaches to that adaptation have predominantly targeted the model's weights. Full fine-tuning is computationally expensive, and parameter-efficient fine-tuning (PEFT) methods such as LoRA (Hu et al., 2021) are lighter but tied to the backbone's internal architecture, requiring per-model choices about where and how to insert trainable modules. Beyond these practical costs, recent empirical studies report mixed effects on accuracy and calibration when weight-space adaptation is applied to TFMs (Rubachev et al., 2025; Tanna et al., 2026). An interpretation is that the synthetic prior carried by the pretrained weights is fragile under further gradient updates: the inductive bias that makes a TFM strong out-of-the-box can be eroded by even modest weight-space changes, leaving these methods with a narrow operating margin between under- and over-adaptation.

We instead adapt the *input*, not the weights. Prior TFM input-space adaptation (BETA (Liu & Ye, 2025)) replaces the original features with a learned encoding—a design shaped by first-generation TabPFN's capacity constraints. With recent TFMs such as TabPFN-2.5/2.6 and TabICLv2 having largely lifted those constraints, the bottleneck is no longer capacity but *alignment*: the frozen model's biases come from a generic synthetic prior that need not match the structure of any specific downstream task. Our framework, *TFM-Retouche* (**T**abular **F**oundation **M**odel **Re**sidual **touch**ed **e**xtension), targets alignment by *nudging* the input rather than replacing it. A near-identity residual adapter applies a small, learned correction to the original input to

[1]Anonymous Institution, Anonymous City, Anonymous Region, Anonymous Country. Correspondence to: Anonymous Author <anon.email@domain.com>.

Preliminary work. Under review by the International Conference on Machine Learning (ICML). Do not distribute.

align it with the inductive biases the frozen model expects, preserving feature dimensionality, treating the pretrained model as a differentiable but otherwise unmodified black box, and requiring no internal modifications.

Our contributions are:

- **The *TFM-Retouche* framework.** A lightweight, configurable input-space residual adapter for frozen TFMs. The adapter is dimension-preserving, architecture-agnostic with respect to the backbone, and supports both classification and regression.

- **End-to-end training through a frozen TFM with a post-training identity guard.** The adapter is trained end-to-end by backpropagation through the frozen TFM forward pass, and a per-fit identity guard scores the adapter path against the unmodified TFM and routes around the adapter at inference whenever adaptation does not improve, bounding the risk of adaptation.

- **Evaluation on TabArena-Lite.** *TabICLv2-Retouche*, the framework instantiated on TabICLv2, is the top-ranked method on the leaderboard with light per-task tuning and ensembling. Appendix E reports component ablations; Appendix F reports head-to-head comparisons against BETA, LoRA, and full SFT on a frozen TabICLv2 backbone; Appendix K extends the evaluation to the TALENT benchmark (Liu et al., 2024) (170 datasets).

## 2. Method

**Problem formulation.** Let $f_\theta$ denote a pretrained tabular foundation model with frozen parameters $\theta$. Let $D^{raw} = \{(x_i^{raw}, y_i)\}_{i=1}^N$ denote a raw downstream dataset and $D = \{(x_i, y_i)\}_{i=1}^N$ its preprocessed counterpart, with $x_i \in \mathbb{R}^d$ and target $y_i$. We seek a lightweight, parameter-efficient adapter $g_\phi : \mathbb{R}^d \to \mathbb{R}^d$, parameterized by $\phi$ with $|\phi| \ll |\theta|$, such that $f_\theta(g_\phi(X))$ outperforms $f_\theta(X^{raw})$. Figure 1 summarizes the framework.

**Positioning.** Three design choices distinguish $g_\phi$ from prior TFM adaptation paradigms:

- **Dimension-preserving.** $g_\phi$ maps $\mathbb{R}^d \to \mathbb{R}^d$, in contrast to BETA's fixed-width MLP encoder (Liu & Ye, 2025), which compresses the input through a 100-dimensional bottleneck.

- **Initialized near identity.** The gate $\alpha$ starts at a small value so that $g_\phi(x) \approx x$ at the start of training, preserving the pretrained prior that weight-space fine-tuning can erode (Rubachev et al., 2025; Tanna et al., 2026).

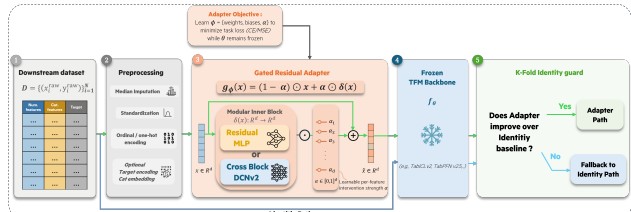

*Figure 1.* **Overview of *TFM-Retouche*.** A preprocessed input $x$ is passed through a gated residual adapter $g_\phi(x) = (1-\alpha) \odot x + \alpha \odot \delta(x)$ before entering a frozen TFM. Only the adapter parameters $\phi$ are trained end-to-end through the frozen backbone $f_\theta$. After training, an identity guard falls back to the unmodified TFM when adaptation does not improve held-out validation performance.

- **Operates only at the input boundary.** The backbone $f_\theta$ is treated as a differentiable but otherwise unmodified black box. This is a more general interface than weight-space PEFT methods such as LoRA (Hu et al., 2021), whose insertion points and ranks must be tailored to each backbone's internal architecture, and it makes *TFM-Retouche* architecture-agnostic with respect to the choice of TFM.

**The adapter.** The core design is a gated adapter with a learnable strength parameter:

$$g_\phi(x) = (1-\alpha) \odot x + \alpha \odot \delta(x), \qquad (1)$$

where $\alpha$ is learnable, $\delta : \mathbb{R}^d \to \mathbb{R}^d$ is a trainable transformation, and $\odot$ is the Hadamard product. This preserves input dimensionality while allowing the adapter to apply a task-specific correction. Equivalently, $g_\phi(x) = x + \alpha \odot (\delta(x) - x)$, so the convex-blend and pure-residual viewpoints coincide; Appendix A gives the derivation. By default, $\alpha \in \mathbb{R}^d$ is a per-channel vector so that each feature has its own intervention strength; a scalar gate $\alpha \in \mathbb{R}$ is also supported and acts as a regularizer when the per-channel form is prone to overfitting. We initialize $\alpha$ to $\alpha_0 = 0.02$, so that $g_\phi(x) \approx x$ at the start of training and identity is recovered exactly when $\alpha = 0$. The motivation is intrinsic to the problem: the frozen backbone already carries a strong default, so we want the TFM to see inputs essentially indistinguishable from those it learned on, departing from the near-identity regime only as the loss signal opens specific channels. This small-initialization choice is similar in spirit to ReZero (Bachlechner et al., 2020) and LayerScale (Touvron et al., 2021).

**Inner block.** The transformation $\delta$ is a stack of $L$ residual layers; depth $L$ is a hyperparameter. We study two interchangeable variants. The *cross block* (DCNv2 (Wang et al., 2021)) uses $x_{l+1} = x_0 \odot (W_l x_l + b_l) + x_l$, with a low-rank factorization $W_l = U_l V_l^\top$ ($U_l, V_l \in \mathbb{R}^{d \times h}$, $h = \lfloor rd \rfloor$). The *residual MLP block* uses $x_{l+1} = x_l + W_l^2 \sigma(W_l^1 x_l + b_l^1) + b_l^2$,

a bottlenecked nonlinear transformation. The two blocks share the outer interface, depth $L$, low-rank/expansion ratio $r$, BatchNorm option, and initialization, and are matched at $O(2dh)$ parameters per layer. We use the cross block as the default and treat the MLP variant as a focused ablation (Appendix E).

**End-to-end training.** We train the adapter parameters end-to-end while keeping the backbone fully frozen. During each epoch, the training data are randomly partitioned into a context set and a query set, matching the in-context-learning setting of the backbone, and the loss is computed only on the query examples. We optimize with AdamW (Loshchilov & Hutter, 2019) using separate parameter groups for matrix weights, biases, and the gate $\alpha$; weight decay applies only to matrices; $\alpha$ is given a larger learning rate (factor of 3 in our TabArena instantiation). Muon (Jordan et al., 2024) is supported as an alternative for matrix parameters. Loss is cross-entropy with optional label smoothing for classification and MSE for regression. The default schedule is a multi-cycle log-spaced cosine schedule (Holzmüller et al., 2025).

**Identity guard.** After training, the identity guard decides per fit whether the trained adapter is used at inference or bypassed. It scores the adapter path $f_\theta(g_\phi(X))$ against the unmodified baseline $f_\theta(X)$ on a held-out validation set using the deployment metric ($1 - \mathrm{AUC}$ for binary, log loss for multiclass, MSE for regression). If the adapter does not improve over the plain baseline by a small tolerance (default 0.5%), the outer estimator routes around the adapter at inference and the prediction is identical to the unmodified TFM. Implementation details (preprocessing, the input-dimension cap with its trainable low-rank projection for high-dimensional datasets, automatic-precision policy, etc.) are deferred to Appendix G.

## 3. Experiments

**Setup.** We evaluate *TFM-Retouche* on TabArena-Lite (Erickson et al., 2025): 51 datasets spanning binary classification, multiclass classification, and regression, ranging from 748 to 150,000 samples and from 5 to 1,777 features. We follow TabArena's 8-fold bagged cross-validation protocol and reuse each fold's per-fold validation split as the external validation set for the identity guard. We instantiate *TFM-Retouche* with frozen TabICLv2 (Qu et al., 2026) as the backbone and denote this instantiation *TabICLv2-Retouche*. TabICLv2 is one of the strongest open TFMs and exposes a forward pass convenient for our end-to-end adaptation setting. The framework is backbone-agnostic by design; extending it to other openly licensed TFMs is left to future work (Section 4). Our primary baseline is the frozen TabICLv2 backbone without adaptation, which isolates the con-

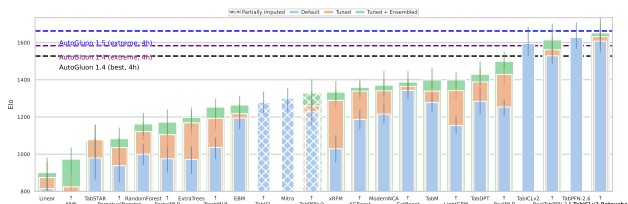

*Figure 2.* TabArena-Lite leaderboard: Elo by method. *TabICLv2-Retouche* (T+E), our framework instantiated on frozen TabICLv2, is the top-ranked method, achieved under a 20× smaller per-dataset HPO budget than every tuned peer (10 random configurations vs. TabArena's standard 200).

tribution of the adapter. We also place *TabICLv2-Retouche* against published TabArena-Lite leaderboard baselines, including RealTabPFN-2.5 and TabPFN-2.6 (Grinsztajn et al., 2026), the standard tabular suite (XGBoost, LightGBM, CatBoost, RealMLP, TabM, TabDPT, ModernNCA, EBM), and additional tree-based and linear baselines; full list in Appendix J. Baseline results are taken directly from the published TabArena leaderboard. We use a lightweight per-dataset random search of 11 configurations (1 default + 10 random trials) over the search space in Appendix H. We report three protocols: **(D)** default configuration, **(T)** the single best per-dataset configuration from the 11-trial search, and **(T+E)** ensembling top configurations across the 8 AutoGluon bag-folds. Note that TabArena's published leaderboard tunes each non-Retouche baseline with 200 random configurations per dataset, so every (T) or (T+E) peer reflects a 20× larger random-search budget than *TabICLv2-Retouche* receives.

**Main results.** Table 1 reports the simplified TabArena-Lite leaderboard. *TabICLv2-Retouche* (T+E) reaches Elo 1651, a +56 gain over the unmodified TabICLv2 base (1595), and takes rank 1 on the leaderboard. The single-best-config variant *TabICLv2-Retouche* (T) follows at Elo 1632 (rank 2, +37 vs. base). The next-best non-Retouche peers, TabPFN-2.6 (D) and RealTabPFN-2.5 (T+E), sit at Elo 1627 and 1614, respectively. *TFM-Retouche*, instantiated on TabICLv2, takes the top two slots of the leaderboard while modifying no pretrained weight and adding only $10^2$ to $10^5$ trainable parameters per dataset.

Figure 2 plots, for each method that ships both a default and a tuned variant, the Elo gain from per-dataset hyperparameter tuning. Unlike baselines whose defaults sit far below their tuned ceiling (e.g. RealMLP, LightGBM, and XGBoost each gain $>170$ Elo from D to T+E), *TabICLv2-Retouche* starts from a strong default at Elo 1608 that already exceeds the unmodified TabICLv2 base, and tuning further lifts it by +43 Elo to 1651 at rank 1. *TabICLv2-Retouche* therefore delivers a useful lift either out of the box or with a small per-task search.

*Table 1.* Simplified TabArena-Lite leaderboard. *TabICLv2-Retouche*'s T and T+E variants use a 20× smaller per-dataset HPO budget than every tuned peer (10 random configurations vs. TabArena's standard 200). Time columns are wall-clock per 1K samples. Bold rows mark *TabICLv2-Retouche* variants. Italic row marks the unmodified TabICLv2.

| Method | Elo (↑) | Norm. (↑) | Avg. rank (↓) | Train [s/K] | Pred. [s/K] |
|---|---|---|---|---|---|
| ***TabICLv2-Retouche* (T+E)** | $1651_{-63,+85}$ | **0.665** | **10.2** | **243.17** | **22.03** |
| ***TabICLv2-Retouche* (T)** | $1632_{-61,+74}$ | **0.659** | **11.0** | **243.17** | **7.33** |
| TabPFN-2.6 (D) | $1627_{-61,+82}$ | 0.643 | 11.3 | 5.75 | 0.60 |
| RealTabPFN-2.5 (T+E) | $1614_{-66,+88}$ | 0.630 | 11.8 | 2059.94 | 9.79 |
| ***TabICLv2-Retouche* (D)** | $1608_{-62,+67}$ | **0.623** | **12.1** | **20.80** | **7.24** |
| TabICLv2 (D) | $1595_{-67,+89}$ | *0.633* | *12.7* | *4.01* | *0.35* |
| RealMLP (T+E) | $1500_{-44,+54}$ | 0.458 | 17.7 | 2791.97 | 13.89 |
| LightGBM (T+E) | $1400_{-41,+41}$ | 0.275 | 24.0 | 416.56 | 2.24 |
| CatBoost (T+E) | $1388_{-45,+60}$ | 0.291 | 24.7 | 1665.53 | 0.56 |
| XGBoost (T+E) | $1360_{-46,+43}$ | 0.231 | 26.7 | 700.96 | 1.44 |
| TabPFNv2 (T+E) | $1328_{-72,+75}$ | 0.312 | 28.9 | 2942.08 | 17.37 |

**Compute trade-off.** On a quality-vs-compute view (Pareto plots in Appendix B, Figures 3 and 4), *TabICLv2-Retouche* (T+E) is the highest-Elo operating point on the training-time Pareto frontier among tuned-and-ensembled methods at 243 s per 1K samples; it dominates every other (T+E) peer on both quality and training time (e.g. RealTabPFN-2.5 (T+E) at 2,060 s, RealMLP (T+E) at 2,792 s). On the inference-time axis, *TabICLv2-Retouche* (T+E) at 22 s/K is the highest-Elo point on the frontier; *TabICLv2-Retouche* (T) lies on the same frontier as a lower-overhead variant. *TabICLv2-Retouche*'s wall-clock numbers were measured on a mix of NVIDIA A10 and A100 GPUs, whereas TabArena's published times for GPU-based baselines were obtained on faster H100s, so the training-time dominance and the inference-time Pareto-frontier position hold *despite* a hardware handicap. More results can be found in the Appendices.

## 4. Conclusion and Limitations

*TFM-Retouche* frames the adaptation of a tabular foundation model as a small, near-identity correction in input space: a learnable gate combined with a lightweight inner block ($10^2$ to $10^5$ trainable parameters per dataset, scaling with input dimensionality), trained end-to-end through the frozen backbone, and gated at deployment by a post-training identity check on held-out validation. No pretrained weight is modified and no TFM internal-architecture choices are required, leaving the backbone a swappable knob rather than a structural commitment. Empirically, *TabICLv2-Retouche* is the rank-1 method on TabArena-Lite at Elo 1651 (T+E, +56 over the frozen base), dominates the training-time Pareto frontier among tuned-and-ensembled methods, and remains on the inference-time Pareto frontier, showing that input-space adaptation can deliver state-of-the-art quality without sacrificing efficiency.

*TFM-Retouche* is intended as a late-stage refinement rather than a replacement for the zero-shot TFM workflow. Mod-

ern TFMs already deliver competitive predictions with no per-dataset training, which makes them well-suited to fast iteration on the upstream decisions (data sources, feature schema, data representation, evaluation, monitoring) where most early-stage data-science effort belongs and where the frozen TFM is hard to beat as a default predictor. Once the upstream pipeline has stabilized, marginal predictive gains start to translate into real-world value, either as raw accuracy or as personalization toward a target distribution, and *TFM-Retouche* provides a small, low-friction mechanism for capturing them. The frozen backbone is left untouched so the existing zero-shot pipeline remains the fallback; the per-dataset adapter trains on a single GPU at adapter-only cost rather than backbone fine-tuning cost; the identity guard (Section 2) routes back to the unmodified TFM on any dataset where adaptation does not improve held-out performance, making the addition strictly opt-in per dataset; and because the adapter is architecture-agnostic with respect to the backbone, the choice of TFM remains a swappable knob, so future stronger TFMs can be incorporated naturally. Together these properties position *TFM-Retouche* as a low-risk, late-stage performance lift after the upstream data and pipeline decisions have stabilized.

Several choices in this paper trade absolute performance for tractability under a constrained compute budget, so the reported results do not represent the full potential of *TFM-Retouche*. We benchmark only on TabICLv2; TabPFN-2.5/2.6 are not yet evaluated owing to license. Several orthogonal ideas from recent tabular work fit naturally on top of this skeleton: richer per-feature numeric encodings such as PLR (Gorishniy et al., 2023), the robust preprocessing pipeline and optimization tricks of RealMLP (Holzmüller et al., 2025), TabM-style BatchEnsemble (Gorishniy et al., 2025) and BETA-style adapter bagging (Liu & Ye, 2025). We expect each to compose cleanly with the framework.

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

## A. Equivalent residual form of the gated adapter

Equation 1 writes the adapter as a convex blend of the raw input $x$ and an inner-block output $\delta(x)$ under a learnable gate $\alpha$. We show here that this is algebraically identical to a pure additive residual on $x$ whose magnitude is controlled by $\alpha$, and that the resulting residual has a clean closed form for both inner blocks of Section 2.

Distributing the gate in Equation 1 gives

$$g_\phi(x) \;=\; (1-\alpha) \odot x + \alpha \odot \delta(x) \;=\; x \;-\; \alpha \odot x \;+\; \alpha \odot \delta(x) \;=\; x \;+\; \alpha \odot \big(\delta(x) - x\big). \tag{2}$$

Defining the inner-block residual

$$r(x) \;:=\; \delta(x) - x, \tag{3}$$

the adapter is therefore a pure additive residual on $x$ with effective gated residual $\alpha \odot r(x)$:

$$g_\phi(x) \;=\; x \;+\; \alpha \odot r(x). \tag{4}$$

Because both inner blocks defined in Section 2 are themselves residual stacks initialized at $x_0 = x$, the inner-block residual $r(x) = \delta(x) - x$ telescopes into a sum of per-layer increments and is available in closed form.

For the cross block, $x_{l+1} = x_l + x_0 \odot (W_l x_l + b_l)$, so

$$r_{\text{cross}}(x) \;=\; \delta_{\text{cross}}(x) - x \;=\; \sum_{l=0}^{L-1} x_0 \odot (W_l x_l + b_l), \tag{5}$$

which makes explicit that the cross-block contribution is a layer-wise sum of input-anchored multiplicative corrections.

For the residual-MLP block, $x_{l+1} = x_l + W_l^2 \sigma(W_l^1 x_l + b_l^1) + b_l^2$, so

$$r_{\text{MLP}}(x) \;=\; \delta_{\text{MLP}}(x) - x \;=\; \sum_{l=0}^{L-1} \Big( W_l^2 \sigma(W_l^1 x_l + b_l^1) + b_l^2 \Big), \tag{6}$$

i.e. a layer-wise sum of bottlenecked nonlinear corrections. In both cases, substituting Equation 5 or Equation 6 into Equation 4 yields an explicit per-channel residual whose overall magnitude is controlled by the learned gate $\alpha$.

## B. Additional headline-run results

This appendix collects two complementary views of the headline cross-block run reported in Section 3: a quality-vs-compute Pareto plot under the alternative *improvability* y-axis, and a Demšar-style critical-difference diagram that ranks the headline run against every non-AutoGluon TabArena-Lite leaderboard baseline.

**Elo vs. training/inference time.** Figure 3 plots Elo against per-thousand-sample training time (left) and inference time (right) for every method on the TabArena-Lite leaderboard. On the training-time axis, *TabICLv2-Retouche* (T+E) sits on the Pareto frontier as the highest-Elo operating point at 243 s per 1K samples; it dominates every other (T+E) peer on both quality and training time. TabPFN-2.6 (D) and the unmodified TabICLv2 (D) cover the fast end of the same frontier. On the inference-time axis, *TabICLv2-Retouche* (T+E) at 22 s/K is the highest-Elo point on the frontier, with *TabICLv2-Retouche* (T) at 7.3 s/K and the unmodified TabICLv2 / TabPFN-2.6 defaults covering the lower-Elo, faster-inference end.

**Improvability vs. compute.** Figure 4 replots the same data with *improvability* on the y-axis (the gap to the best per-task method, averaged across tasks; lower is better (Erickson et al., 2025)). The Pareto frontier conclusion is invariant to this re-axis: *TabICLv2-Retouche* (T+E) retains its position as the highest-quality (lowest-improvability) operating point on the training-time Pareto frontier among tuned-and-ensembled methods, and remains on the inference-time frontier.

**Critical-difference diagram.** Figure 5 consolidates per-task ranks across TabArena-Lite into a single Demšar-style critical-difference diagram (Friedman test with Nemenyi post-hoc, $\alpha = 0.05$) over the headline *TabICLv2-Retouche* (cross) (T+E) run and every non-AutoGluon TabArena-Lite leaderboard method at its strongest protocol, for a total of 24 entries.

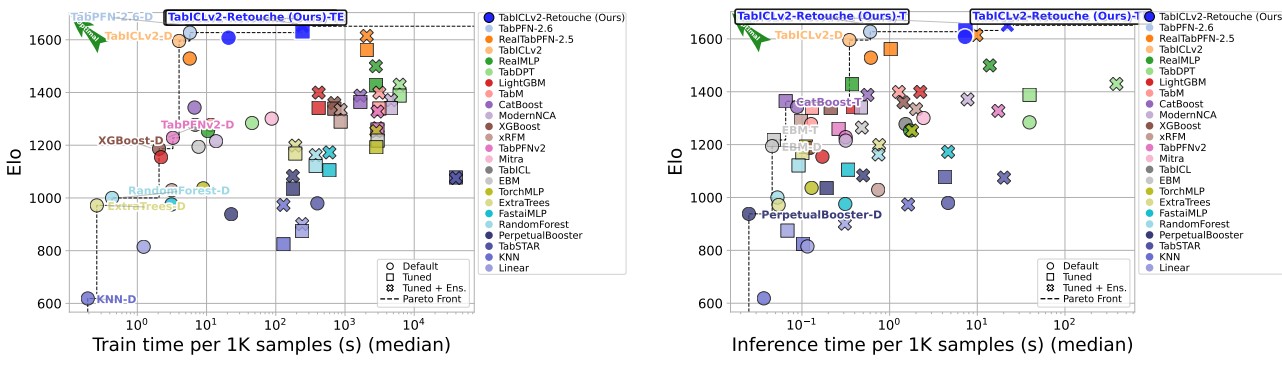

*(a)* Elo vs. training time per 1K samples.

*(b)* Elo vs. inference time per 1K samples.

*Figure 3.* Quality vs. compute trade-off on TabArena-Lite. Upper-left is better; markers denote leaderboard methods. *TabICLv2-Retouche* (T+E) is the highest-Elo point on the training-time Pareto frontier (a) and on the inference-time frontier (b).

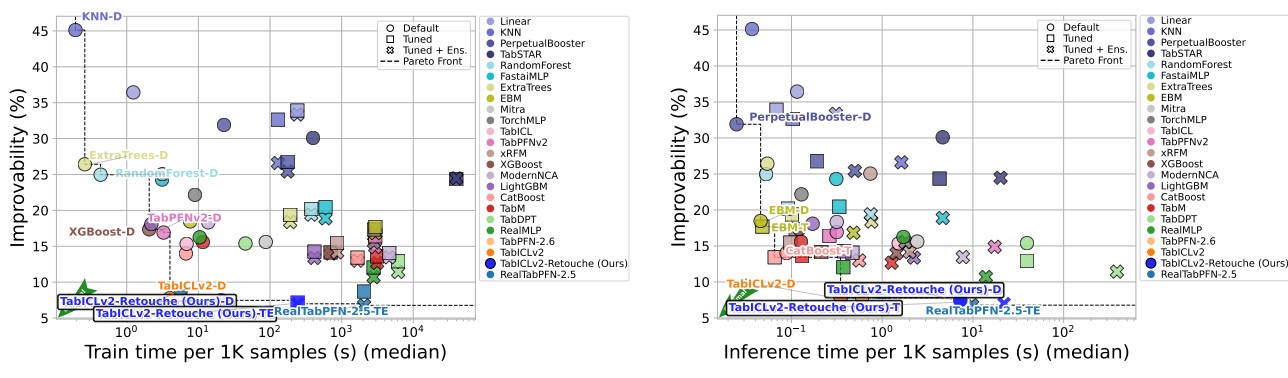

*(a)* Improvability vs. training time per 1K samples.

*(b)* Improvability vs. inference time per 1K samples.

*Figure 4.* Quality vs. compute trade-off on TabArena-Lite: improvability vs. training time (a) and improvability vs. inference time (b). Lower-left is better; markers denote leaderboard methods.

*TabICLv2-Retouche* (T+E) achieves the lowest mean rank, ahead of TabPFN-2.6 (D), RealTabPFN-2.5 (T+E), and the unmodified TabICLv2 (D). It sits in the top significance group, which spans TabPFN-2.6 (D), RealTabPFN-2.5 (T+E), TabICLv2 (D), RealMLP (T+E), TabDPT (T+E), and TabM (T+E); within this group the pairwise rank differences are not statistically significant at $\alpha = 0.05$, so *TabICLv2-Retouche*'s top position is best read as "tied at the top" rather than as a strict separation from the strongest baselines. The remaining tabular methods (gradient-boosted trees, the legacy TFMs, and the simpler MLP and tree baselines) form a long tail of partially overlapping significance groups extending out to KNN and Linear at the bottom of the diagram.

## C. Analysis: where does the adapter intervene?

**Identity-guard activation rate.** The identity guard (Section 2) selects the adapted path on $67.7\%$ of runs ($3{,}040/4{,}488$, counted at the dataset $\times$ config $\times$ fold level in the headline cross-block batch), where the adapter improves over the unmodified TabICLv2 base on the held-out AutoGluon bag-fold validation set; on the remaining $32.3\%$ ($1{,}448/4{,}488$) the guard routes back to the base, ensuring the adapter is never deployed when it would not help. Because the validation set is held out from the per-fold trainer, this is a clean estimate of how often the adapter delivers a genuine gain.

**Per-dataset fallback distribution.** Figure 6 plots the fallback rate per dataset. For each dataset we restrict to the single configuration that minimizes the held-out AutoGluon bag-fold validation error and report the fraction of its 8 AutoGluon bag-folds on which the guard fired. Across the 51 datasets the mean per-dataset fallback rate is $25.2\%$ (median $25.0\%$, interquartile range $[12.5\%, 37.5\%]$), with $12/51$ datasets at $0\%$ (the trained adapter wins on every fold), no dataset at $100\%$, and a single dataset (MIC, multiclass with 111 features) at the maximum of $75\%$.

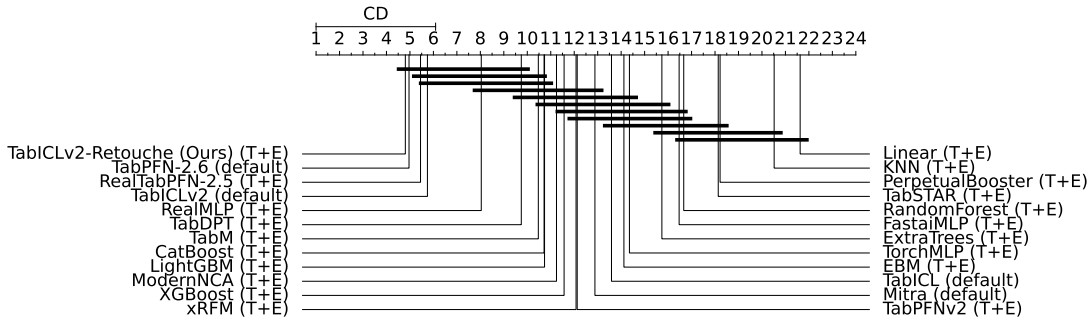

*Figure 5.* Critical-difference diagram (Friedman test with Nemenyi post-hoc, $\alpha = 0.05$) over per-task ranks on TabArena-Lite, covering the headline *TabICLv2-Retouche* (cross) (T+E) run and every non-AutoGluon TabArena-Lite leaderboard method at its strongest protocol (24 entries total). Mean rank is on the horizontal axis (lower is better); horizontal bars connect methods whose pairwise rank difference is not statistically significant.

The pattern is consistent with TabICLv2 being a strong starting point. Regression datasets are the most adapter-friendly (mean fallback 14.4%, median 0% across 13 datasets; 7/13 at $\leq 5\%$), while binary classification ($n\!=\!30$, mean 29.2%) and multiclass classification ($n\!=\!8$, mean 28.1%) show more guard activity.

## D. Cross-block weight inspection on a financial dataset

The cross block in *TabICLv2-Retouche* computes

$$x_{l+1} = x_0 \odot (W_l\, x_l + b_l) + x_l,$$

where $\odot$ denotes the Hadamard product. Component $i$ of the layer-$(l+1)$ output is a sum of terms $x_0[i] \cdot W_l[i,j] \cdot x_l[j]$, so $W_l[i,j]$ is the explicit coefficient of the multiplicative interaction $x_0[i] \cdot x_l[j]$. After fitting we can read these weights off and ask which feature pairs the adapter places weight on.

To estimate cumulative effects across $L$ layers, we model the cross block as a vector-valued map $\mathrm{cross} : \mathbb{R}^d \to \mathbb{R}^d$, and we collapse it to the scalar function $f(x) = \sum_k \mathrm{cross}(x)_k$ by summing its $d$ output channels with equal weight, so that the second derivative of $f$ is a single $d \times d$ matrix. We then read off the symmetrised numerical Hessian $H[i,j] = \frac{1}{2}\big(\partial^2 f/\partial x_i\, \partial x_j + \partial^2 f/\partial x_j\, \partial x_i\big)$ via autograd, evaluated at the test-set column mean (the per-feature average of the preprocessed test inputs). Each entry $H[i,j]$ aggregates the multiplicative coupling between features $i$ and $j$ across all cross layers and all output channels, including BatchNorm rescaling and the inner activation between the low-rank factors $V_l\, U_l$ when one is present. We work with $H$ rather than the analytical aggregate $\sum_l \frac{1}{2}(W_l + W_l^\top)$ because this dataset's inner block can be low-rank with a ReLU between $V_l$ and $U_l$.

We analyze $|H|$ for the first bag-member of the (T) version of *TabICLv2-Retouche* (the per-dataset best-tuned configuration; see Section 3) on taiwanese_bankruptcy_prediction (TabArena-Lite, ID 363706, binary classification, 94 numeric financial-ratio features). Among the 15 largest off-diagonal entries of $|H|$ (Figure 7), 2 involve feature pairs whose two ratios have both appeared as separate regressors in named bankruptcy-prediction studies.[1] We list those alignments below without claiming the network has identified the underlying generative process:

- Debt_Ratio_Percent × Total_Assets_to_GNP_Price. Leverage interacted with a real-deflated firm-size measure. Both ratios appear directly as regressors in the O-score logit model of Ohlson (1980). Size and leverage also enter the discrete-time hazard models of Shumway (2001) and Chava & Jarrow (2004) as separate regressors.

- ROA_C_Before_Interest_Depreciation × Working_Capital_to_Equity. A profitability and working-capital pair within the family of variables used by the discriminant Z-score of Altman (1968).

---

[1]Explicit interaction terms have been studied in this literature: Laitinen & Laitinen (2000) add second-order and pairwise products of $\mathrm{Cash}/TA$, $\mathrm{CashFlow}/TA$, and $\mathrm{Equity}/TA$ to a logit model and report improved accuracy 1–2 years before bankruptcy. The specific pairs probed there differ from those highlighted by the cross block here.

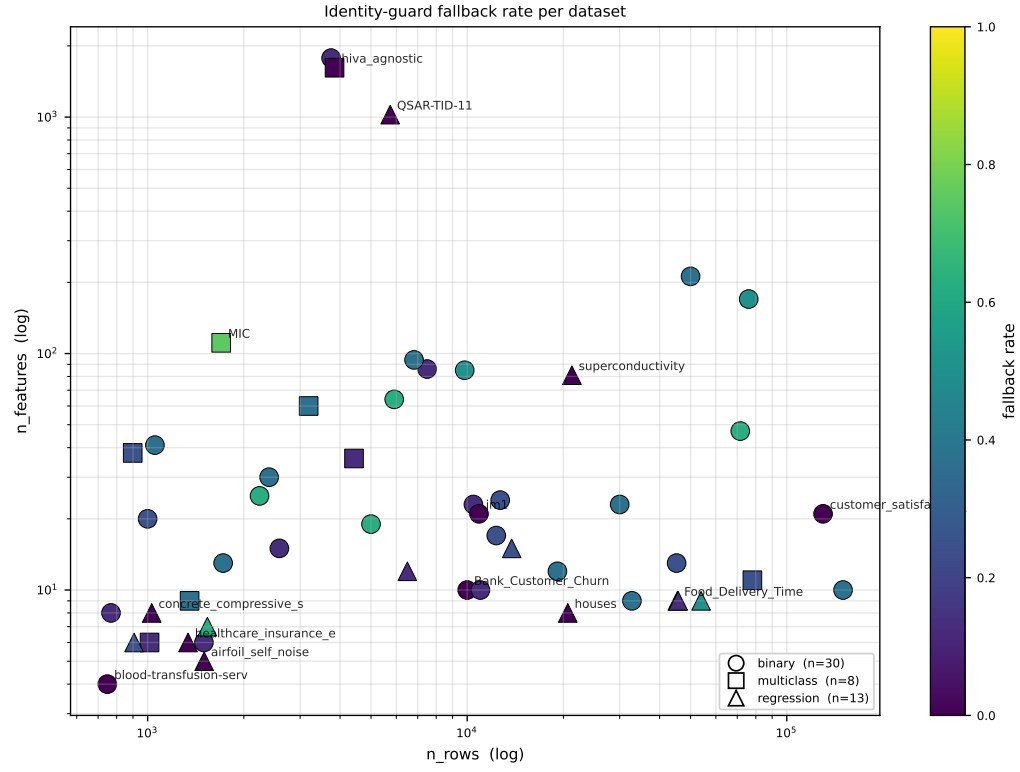

*Figure 6.* Per-dataset identity-guard fallback rate for the headline cross-block run of *TabICLv2-Retouche* on TabArena-Lite. Each marker is one dataset, positioned by its number of rows (horizontal, log scale) and number of features (vertical, log scale). The color encodes the fraction of the 8 AutoGluon bag-folds on which the guard routed back to the unmodified TabICLv2 base, restricted to the single configuration that minimizes the held-out AutoGluon bag-fold validation error per dataset. Marker shape encodes problem type (circle: binary, square: multiclass, triangle: regression). Datasets at the extremes (fallback rate $\geq 0.75$ or $\leq 0.05$) are annotated.

The remaining top-15 entries involve narrower accounting variables, for example Allocation_Rate_Per_Person or Persistent_EPS_Last_4_Seasons, for which we do not have a specific reference and which we report without further interpretation.

**Caveats.**

- The dataset has many near-duplicate ratios (six profitability variants, five leverage variants, three liquidity variants), and the cross block carries low rank by construction. Under such multicollinearity, a low-rank fit is expected to select one proxy per economic concept rather than all of them: the textbook pair Quick_Ratio × Interest_Expense_Ratio (a liquidity-versus-debt-burden interaction) appears here with small magnitude, while the closely related Cash_to_Total_Assets × Interest_Expense_Ratio appears among the largest entries of $|H|$. The bullet list above should therefore not be read as evidence that the network has identified those specific documented variables. Substituting alternative proxies under multicollinearity is expected behavior for any low-rank fit and does not bear on which proxy is causally preferable.

- This is a single illustrative dataset with named features. **We do not extrapolate** the alignments observed here to other datasets in TabArena-Lite, several of which use anonymized feature names that preclude domain interpretation. The intent of this appendix is narrow: the cross block's weights are inspectable, and on this dataset 2 of the 15 largest entries of $|H|$ correspond to feature pairs whose two ratios both appear as separate regressors in named bankruptcy-prediction studies.

# E. Ablations

We conducted ablation studies to understand the impact of each element in the design of *TFM-Retouche*.

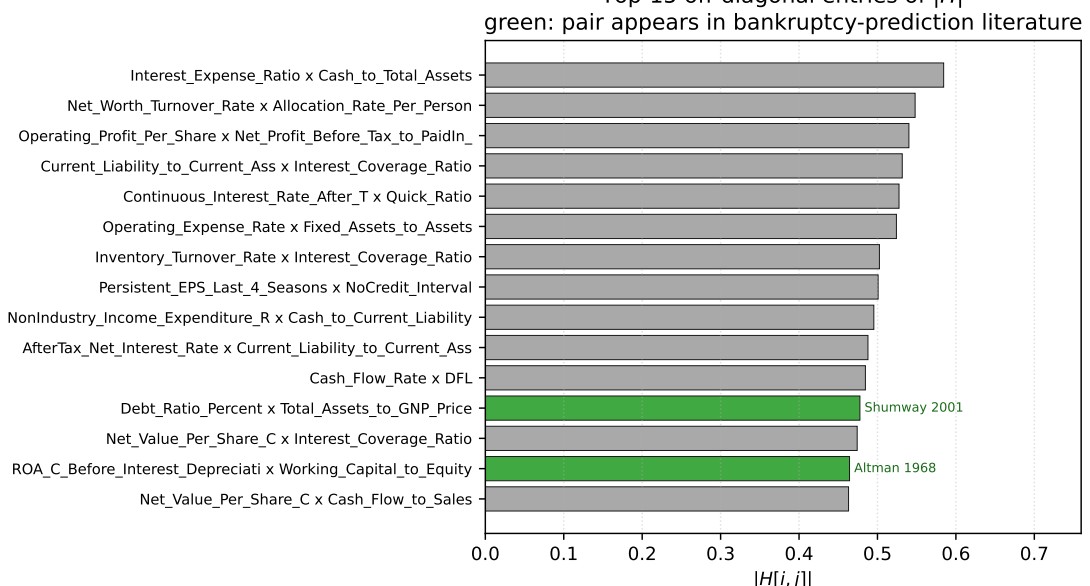

*Figure 7.* Magnitudes of the 15 largest off-diagonal entries of $|H|$ for the first bag-member of *TabICLv2-Retouche* (T) on tai-wanese_bankruptcy_prediction (TabArena-Lite ID 363706), sorted with the largest at the top. Pairs whose two ratios both appear as separate regressors in a named bankruptcy-prediction study (Altman, 1968; Shumway, 2001; Chava & Jarrow, 2004; Ohlson, 1980) are coloured green; the remaining pairs are grey.

**Block type (cross vs. MLP vs. identity)**: We compare the DCNv2 cross block (our default), the residual-bottleneck MLP block, and identity (no adapter, equivalent to the frozen TabICLv2 backbone) at matched seeds and an otherwise identical configuration grid: the same 11 hyperparameter draws used in the main results (Appendix H). The cross-block run is the headline batch reported in Section 3; the paired MLP-block reuses every other knob unchanged.

**Training ablation (Random Adapter)**: We observed that the identity guard (Section 2) routed back to the unmodified frozen TabICLv2 on a sizable fraction of runs: 32.3% at the run level ($1{,}448/4{,}488 = 51$ datasets $\times$ 11 configurations $\times$ 8 folds) in the headline cross-block batch, where the trained adapter did not improve over the base on the held-out fold. This raised a natural concern: does the lift come from end-to-end training of the adapter, or could it instead be explained by randomness and the 8-fold ensembling? To isolate the contribution of training, we re-run *TabICLv2-Retouche* with the adapter randomly initialized at the beginning of training and never updated. All other settings are held identical to the headline run; only the gradient updates to the adapter are removed.

**Gate and identity-guard variants**: The block-type and Random Adapter ablations probe two of the three safety knobs of *TabICLv2-Retouche*: the choice of inner block $\delta$ and end-to-end optimization of the adapter parameters. We complement them with four further runs that probe the trainable per-channel gate $\alpha$ (Section 2) and the identity guard (Section 2).

All the *TabICLv2-Retouche* variants share the headline 11-configurations and the 8-fold AutoGluon bagging protocol; each variant modifies a single knob:

- **[cross]** Headline cross-block run.

- **[mlp]** Block-type ablation: residual-bottleneck MLP $\delta$ in place of the DCNv2 cross block.

- **[cross, alpha_init+0.5]** Gate-init ablation: per-config $\alpha$ initialization shifted by $+0.5$ from the headline default; $\alpha$ remains trainable.

- **[cross, alpha= 1]** Strict-freeze ablation: $\alpha$ initialized at 1 and held fixed; the adapter reduces to $f_\theta \circ \delta$ with no gated residual.

- **TabICLv2**: the unmodified frozen TabICLv2 base run at its default (D) configuration: no adapter, no gate, no identity guard. Serves as the no-adapter floor against which each *TabICLv2-Retouche* variant is measured.

**Pairwise win rate on TabArena-Lite**

Opponent (column loses)

*Figure 8.* Pairwise win rates (%) among *TabICLv2-Retouche* variants (T+E) and the unmodified TabICLv2 (D) on TabArena-Lite. Cell $(i, j)$ reports the percentage of datasets on which row method $i$ outperforms column method $j$.

- [cross, random] Training ablation (Random Adapter, above): $\delta$ randomly initialized and never updated.

- [cross, no-guard] Guard ablation: identity guard disabled, so the trained adapter is always deployed.

- [cross, alpha= 1, no-guard] Combined ablation: $\alpha$ frozen at 1 and guard disabled.

We report pairwise win rates rather than Elo here: the pool consists of many variants of one method, so an Elo fit would inflate the headline rating with easy wins against its own degenerate ablations. A pairwise win rate, by contrast, is a direct head-to-head statistic that does not depend on the rest of the pool. Figure 8 reports pairwise win rates (%) among the *TabICLv2-Retouche* variants (T+E) and the unmodified TabICLv2 base (D) on TabArena-Lite. Cell $(i, j)$ is the percentage of datasets on which row method $i$ outperforms column method $j$. The headline cross-block run leads the matrix, beating every other entry by at least 57–43. The cross-vs-MLP edge (57–43) is the narrowest non-degenerate gap. Both gate ablations cost the same: [cross, alpha_init+0.5] and [cross, alpha= 1] each lose to the headline at 33–67, indicating that the small near-identity initialization and the trainability of $\alpha$ contribute to the performance of the headline. The training ablation [cross, random] loses at 31–69, and the guard ablation [cross, no-guard] loses at the same 31–69. The doubly-ablated variant [cross, alpha= 1, no-guard] wins only 8–16% across all opponents. Without either safety mechanism the trained adapter actively hurts on average.

The relative performance of the variants to the unmodified TabICLv2 base sharpens the picture. First, the headline cross-block lifts the base by 61–39 in pairwise wins and the MLP block by 55–45, so both inner blocks deliver a lift over no-adapter. Second, the strict-freeze ablation [cross, alpha= 1] essentially *ties* the base at 51–49: with the gate frozen at 1 and the identity guard intact, the framework is no better than running TabICLv2 alone, so the small-init trainable $\alpha$ is what allows the lift to materialize on top of the guard.

The most striking cell is [cross, random] vs. [cross, no-guard]: the untrained-but-guarded random adapter wins 65–35 over the trained-but-unguarded adapter. On average across TabArena-Lite, the safety floor of the identity guard contributes more to the headline lift than end-to-end optimization of the adapter does.

The pairwise comparisons support a layered reading of *TabICLv2-Retouche*'s components:

- The identity guard is the single most important component: it provides more pairwise wins than training the adapter end-to-end ([cross, random] beats [cross, no-guard] 65–35), and removing it drives the trained adapter *below* the unmodified TabICLv2 base (39–61).

- The trainable gated residual is the second-most important: with the guard intact, freezing $\alpha$ at 1 erases the framework lift and only *ties* the base (51–49), and shifting the initialization by $+0.5$ keeps a 59–41 edge over the base while losing 33–67 to the headline default.

- Removing both safety mechanisms simultaneously is catastrophic: [cross, alpha$=1$, no-guard] wins only 8–16% across all opponents and loses to the unmodified base 16–84, validating the framework prescription that the gated residual and the identity guard are jointly load-bearing rather than redundant safeguards.

## F. Head-to-head: external adaptation baselines

The Ablations appendix (Appendix E) probes *TabICLv2-Retouche*'s internal design choices. Here we instead compare the framework against the three contemporary adaptation paradigms applicable to a frozen TabICLv2 stack: weight-space full supervised fine-tuning, PEFT (LoRA), and input-space encoder-and-bagging (BETA).

All four trained methods share the headline 11-configurations HPO grid (Appendix H) and the 8-fold AutoGluon bagging protocol; each baseline is run via the public reference implementation indicated below:

- TabICLv2-Retouche: the headline cross-block run reported in Section 3.

- TabICLv2: unmodified frozen TabICLv2 (Qu et al., 2026) at its default configuration.

- TabICLv2-LoRA (TabTune): we vendor TabTune (Tanna et al., 2025), reusing only its `apply_tabular_lora` helper to inject LoRA adapters into the TabICLv2 stack. The surrounding training loop, optimizer setup, batching, ICL context/query split, early stopping, and identity guard are the iso-Retouche pipeline used by the headline batch (we do not invoke TabTune's own `simple_sft` or `meta-learning` fine-tuning entry points). All backbone parameters are explicitly frozen before injection so that only the LoRA `lora_A`/`lora_B` weights are trainable.

- TabICLv2-SFT (TabTune): every backbone parameter is unfrozen and updated end-to-end. The TabTune library is not invoked here; we run the iso-Retouche training loop on the unfrozen backbone with the SFT-style hyperparameters reported in TabTune's reference recipe (Tanna et al., 2025) (Adam, learning rate $10^{-5}$, cross-entropy). The "(TabTune)" tag therefore identifies the recipe, not the executed code path: the per-dataset HPO grid, the bagging protocol, and the identity guard are the same as for the LoRA arm and the headline batch.

- BETA-TabICLv2: the BETA recipe (Liu & Ye, 2025) (a two-layer fixed-width MLP encoder placed before the frozen TFM and trained end-to-end) ported to the TabICLv2 backbone. The published BETA implementation targets first-generation TabPFN; we re-host its encoder/training loop on top of TabICLv2 so the head-to-head is iso-backbone with the other rows. Our wrapper currently runs single-forward inference rather than the published 16-encoder bootstrap bagging at predict time, so the BETA-TabICLv2 numbers reported here should be read as a lower bound on what the published recipe would deliver on TabICLv2.

The matrix in Figure 9 reports pairwise win rates (%) on the TabArena-Lite classification subset (Erickson et al. (2025)'s 38-task classification block, minus two tasks: Bioresponse (363620) and hiva_agnostic (363677), both excluded because the gradient-path baselines—LoRA and full SFT—could not complete on them within available GPU memory (A100); the comparison scope is therefore 36 datasets. *TabICLv2-Retouche* itself completes on all 38 classification tasks. The exclusion here is solely to keep the head-to-head matrix iso-protocol across all four trained methods.

The matrix supports four findings.

- *TabICLv2-Retouche* is the only adaptation strategy that beats the unmodified TabICLv2 base. It wins 58–42 over the base, 61–39 over both LoRA and SFT, and 92–8 over BETA. Among the five rows it is the only one with strictly above-50% win rates against every other entry in the comparison pool.

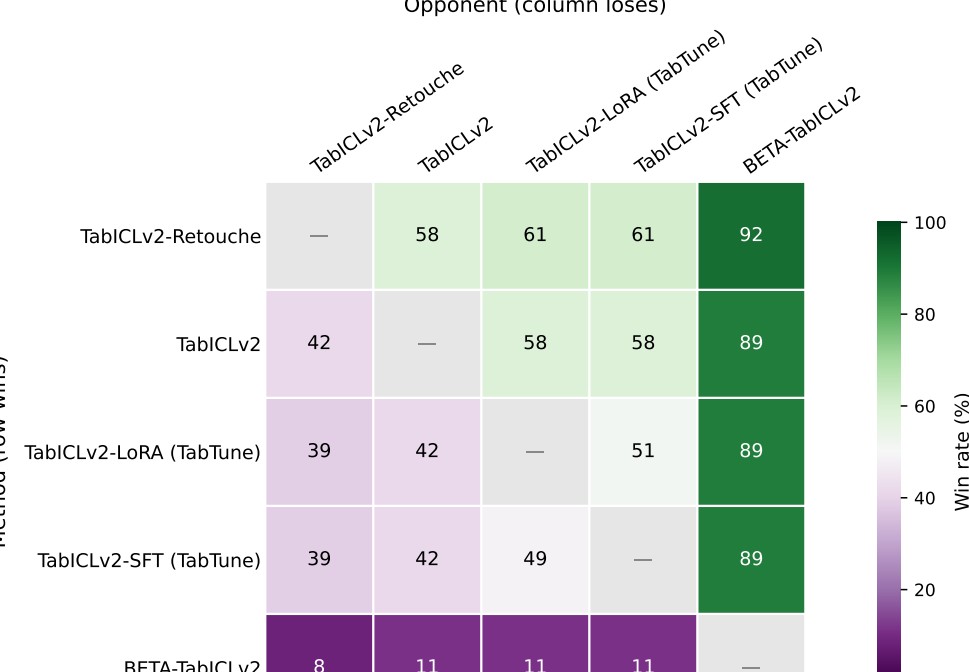

*Figure 9.* Pairwise win rates (%) for the four iso-backbone adaptation paradigms and the unmodified base on the TabArena-Lite classification subset (36 datasets, two datasets excluded for baseline OOM).

- Weight-space adaptation (both LoRA and full SFT) does not improve over zero-shot TabICLv2 on this subset. LoRA loses to the base 42–58 and full SFT loses by the same 42–58. With our matched 11-configuration HPO budget and TabArena-Lite's per-task scale, neither parameter-efficient nor full weight-space tuning yields a measurable gain over the frozen backbone, consistent with the cross-model finding of Tanna et al. (2026) that weight-space gains on already-strong zero-shot TFMs are model- and data-dependent.

- LoRA and full SFT are essentially indistinguishable on this subset (51–49 in either direction). Holding the backbone, the HPO grid, and the bagging protocol fixed, the two weight-space recipes converge to the same per-task error distribution, so the choice between PEFT and full fine-tuning is dominated by their compute cost (full SFT carries the optimizer state for every backbone parameter; LoRA does not) rather than by accuracy.

- Replacing the input rather than nudging it actively hurts when the backbone is already strong. BETA-TabICLv2 loses to the bare base 11–89. The BETA encoder was designed against first-generation TabPFN's capacity constraints, where compressing the input was practically necessary to keep the in-context window tractable. Mounted on TabICLv2, where that constraint no longer binds, the same fixed-width MLP encoder destroys information that the backbone could otherwise consume directly. *TabICLv2-Retouche*'s near-identity, dimension-preserving residual nudge avoids this failure mode by leaving the original feature representation intact and intervening only in proportion to a learnable per-channel gate.

Taken together, the matrix supports the central positioning of the framework: among the four adaptation paradigms compatible with a strong frozen TFM, the input-space residual nudge is the only one that delivers a measurable lift over the zero-shot backbone on TabArena-Lite classification. The bottleneck has indeed shifted from capacity (where BETA's encoder helped on TabPFN-v1) to alignment (where every weight-space path tested here ties or loses to the base, and only *TabICLv2-Retouche* clears it).

## G. Implementation details

**Preprocessing.** We map all input features to a common continuous representation before applying the adapter. Numeric features are median-imputed and standardized. Categorical features are encoded as ordinal integers and then standardized to the same scale. This choice keeps the input dimensionality fixed and ensures that all coordinates enter the adapter on a comparable scale, which is particularly important for the cross block, whose Hadamard products are sensitive to feature magnitudes.

We also consider different options for preprocessing:

- Robust encoder (Holzmüller et al., 2025) and numerical encoding, such as PLR, for numeric features (Gorishniy et al., 2023).

- Mixed preprocessing variant in which low-cardinality categorical features (at most 8 levels) are one-hot encoded, while higher-cardinality features remain ordinal-encoded and standardized (Holzmüller et al., 2025).

- Target embedding or learnable Categorical embedding for categorical features (Holzmüller et al., 2025).

These alternatives can encode numerical features better, and/or expose finer-grained categorical structure, but at the cost of increasing dimensionality and weakening the "close-to-raw-input" character of the overall pipeline. Unless otherwise stated, the experiments reported in this paper use the default variant (standard scaler for numerical features and ordinal encoding for categorical features).

**Other.** To keep adaptation compatible with the backbone across a wide range of input dimensionalities, we cap the feature dimension seen by the TFM at 500. For datasets with $d > 500$, we therefore insert a fixed-rank trainable linear projection between the adapter and the backbone; this projection is orthogonally initialized and serves only to meet the backbone's effective input budget, not to alter the adapter formulation itself. We also support Truncated-SVD as a non-learned alternative. We additionally reshuffle the context/query split at each epoch. Finally, the frozen TFM forward follows the same automatic precision policy as TabICL: fp32 is used on smaller problems ($n < 1024$, $d < 60$), fp16 autocast on medium and large problems, and FA3 attention on sufficiently large inputs ($n \geq 10{,}240$). These choices are purely computational and leave the adapter architecture unchanged.

## H. Hyperparameter search space

For each TabArena-Lite dataset, we evaluate 11 configurations: one default configuration plus 10 random draws from the search space below. The same 11 configurations are evaluated on every dataset; they are sampled once with a fixed seed and reused across the headline batch and its companion batches (random adapter, no-fallback ablation, TabTune-LoRA, TabTune-SFT, and BETA head-to-heads, etc.). Numeric parameters are drawn from continuous ranges either uniformly or log-uniformly; categorical parameters are drawn uniformly from their listed sets; integer parameters are drawn uniformly over the listed inclusive range. The default values in Table 2 are those of the first configuration of the batch reported in Section 3.

## I. Computational details

**Compute platform.** All *TabICLv2-Retouche* runs reported in this paper were submitted as Databricks job runs on Microsoft Azure. Two Azure GPU SKUs were used:

- **A10:** `Standard_NV36ads_A10_v5` (1× NVIDIA A10, 24 GB HBM).

- **A100:** `Standard_NC24ads_A100_v4` (1× NVIDIA A100, 80 GB HBM).

**Sharding.** The headline batch reported in Section 3 was split across 17 Databricks job runs: 6 A10 job clusters each handling a 1/6 shard of the lighter TabArena-Lite tasks via the official sharded suite, plus 11 A100 job clusters each pinned to a single heavier task (task IDs 363616, 363620, 363628, 363630, 363631, 363673, 363677, 363683, 363697, 363699, 363705). Each shard ran the 11 HPO configurations (one default plus 10 random draws; full search space in Appendix H) over the standard 8-fold AutoGluon bagging protocol (Erickson et al., 2025), for a total of $51 \times 11 \times 8 = 4{,}488$ end-to-end fits per batch.

*Table 2.* Hyperparameter search space for *TabICLv2-Retouche*. Defaults match the first of the 11 configurations evaluated in the headline cross-block run (Section 3).

| Hyperparameter | Distribution | Range / choices | Default |
|---|---|---|---|
| *Adapter architecture* | | | |
| num_layers | uniform choice | $\{1, 2\}$ | 2 |
| low_rank_ratio[a] | uniform / full-rank | $[0.1, 0.5] \cup \{\text{None}\}$ | 0.25 |
| hidden_dim | fixed | 64 | 64 |
| use_batch_norm | uniform choice | $\{\text{False, True}\}$ | True |
| alpha_init | log-uniform | $[0.01, 0.1]$ | 0.02 |
| alpha_shape | uniform choice | $\{\text{per-channel, global}\}$ | per-channel |
| gate_lr_factor | log-uniform | $[2.0, 10.0]$ | 3.0 |
| block_type[b] | fixed (cross-arm) | cross | cross |
| *Optimization* | | | |
| optimizer | uniform choice | $\{\text{AdamW, Muon}\}$ | AdamW |
| lr | log-uniform | $[10^{-3}, 1.5 \times 10^{-2}]$ | $5 \times 10^{-3}$ |
| weight_decay | log-uniform | $[10^{-3}, 5 \times 10^{-2}]$ | $3 \times 10^{-3}$ |
| max_grad_norm | log-uniform | $[1.0, 5.0]$ | 2.0 |
| label_smoothing | uniform | $[0.05, 0.30]$ | 0.15 |
| beta2 | uniform | $[0.95, 0.99]$ | 0.97 |
| *Training schedule* | | | |
| epochs | integer uniform | $\{100, \ldots, 200\}$ | 150 |
| patience | integer uniform | $\{10, \ldots, 15\}$ | 10 |
| lr_schedule | uniform choice | $\{\text{cosine, coslog4}\}$ | coslog4 |
| *Initialization and preprocessing* | | | |
| weight_init | uniform choice | $\{\text{xavier-normal, small-normal}\}$ | small-normal |
| activation | uniform choice | $\{\text{None, ReLU}\}$ | None |
| preprocessor | uniform choice | $\{\text{ordinal-scaled, onehot-ordinal}\}$ | ordinal-scaled |

[a] low_rank_ratio applies only when block_type is cross; sampled as full rank (None) with probability $1/3$ and otherwise uniformly from $[0.1, 0.5]$. [b] The result reported in Section 3 has block_type fixed to cross; the paired $\{\text{cross, mlp}\}$ sweep is the block-type ablation in Appendix E.

**Hardware-asymmetry caveat.** Time-per-1K-samples figures reported on the Pareto plots (Figures 3 and 4) were measured on this A10/A100 mix, while published TabArena baselines for GPU methods were measured on faster H100s. As discussed in Section 3, this biases the comparison *against TabICLv2-Retouche*; under matched hardware its wall-clock numbers would only shift further toward the cheap-time end of the plots.

# J. Full TabArena-Lite leaderboard

For completeness, Table 3 reports the complete TabArena-Lite leaderboard for the headline cross-block run of *TabICLv2-Retouche* (Section 3), covering all baseline methods at all three evaluation protocols: (D) default configuration, (T) tuned over 10 random HPO configurations, and (T+E) tuned and ensembled across the 8 AutoGluon bagging folds. Methods are ranked by Elo with bootstrap $95\%$ confidence intervals shown as $_{-l, +u}$ subscripts. Within each column, the best entry is rendered in **bold** and the second best in *italic*. Train and predict times are reported per 1,000 samples; as discussed in Appendix I, our runs use a mix of A10 and A100 GPUs while published baselines were measured on H100, which biases the time comparison *against TabICLv2-Retouche*.

*Table 3.* Full TabArena-Lite leaderboard for the headline cross-block run of *TabICLv2-Retouche* (Section 3). Ranked by Elo. Best per column in **bold**, second in *italic*. Train/predict times are per 1,000 samples.

| Model | Elo ($\uparrow$) | Norm. score ($\uparrow$) | Avg. rank ($\downarrow$) | Harm. mean rank ($\downarrow$) | #wins ($\uparrow$) | Improva-bility ($\downarrow$) | Train time per 1K [s] | Predict time per 1K [s] |
|---|---|---|---|---|---|---|---|---|
| AutoGluon 1.5 (extreme, 4h) | $\mathbf{1662}_{-84,+105}$ | **0.710** | **9.8** | **3.2** | **9.0** | **5.1%** | 293.65 | 4.36 |
| *TabICLv2-Retouche* (Ours) (T+E) | $1651_{-63,+85}$ | *0.665* | *10.2* | 5.0 | 1.3 | 7.0% | 243.17 | 22.03 |
| *TabICLv2-Retouche* (Ours) (T) | $1632_{-61,+74}$ | 0.659 | 11.0 | 4.3 | 5.3 | 7.1% | 243.17 | 7.33 |

*(continues on next page)*

*(continued)*

| Model | Elo (↑) | Norm. score (↑) | Avg. rank (↓) | Harm. mean rank (↓) | #wins (↑) | Improva- bility (↓) | Train time per 1K [s] | Predict time per 1K [s] |
|---|---|---|---|---|---|---|---|---|
| TabPFN-2.6 (D) | $1627_{-61,+82}$ | 0.643 | 11.3 | 5.3 | 3.0 | 7.8% | 5.75 | 0.60 |
| RealTabPFN-2.5 (T+E) | $1614_{-66,+88}$ | 0.630 | 11.8 | 4.2 | 4.0 | *6.8%* | 2059.94 | 9.79 |
| *TabICLv2-Retouche* (Ours) (D) | $1608_{-62,+67}$ | 0.623 | 12.1 | 5.9 | 1.3 | 7.5% | 20.80 | 7.24 |
| TabICLv2 (D) | $1595_{-67,+89}$ | 0.633 | 12.7 | *4.0* | *6.0* | 7.8% | 4.01 | 0.35 |
| AutoGluon 1.4 (extreme, 4h) | $1583_{-68,+82}$ | 0.605 | 13.3 | 5.8 | 2.0 | 8.2% | 556.15 | 6.31 |
| RealTabPFN-2.5 (T) | $1561_{-64,+74}$ | 0.556 | 14.3 | 6.9 | 2.0 | 8.7% | 2059.94 | 1.03 |
| RealTabPFN-2.5 (D) | $1529_{-47,+66}$ | 0.508 | 16.0 | 9.0 | 0.0 | 8.9% | 5.71 | 0.61 |
| AutoGluon 1.4 (best, 4h) | $1528_{-59,+56}$ | 0.502 | 16.1 | 7.7 | 1.0 | 9.7% | 1754.94 | 1.77 |
| RealMLP (T+E) | $1500_{-44,+54}$ | 0.458 | 17.7 | 10.6 | 1.0 | 10.7% | 2791.97 | 13.89 |
| TabDPT (T+E) | $1430_{-59,+69}$ | 0.403 | 22.0 | 7.4 | 3.0 | 11.4% | 6154.73 | 386.17 |
| RealMLP (T) | $1428_{-59,+52}$ | 0.378 | 22.0 | 12.8 | 0.0 | 12.0% | 2791.97 | 0.37 |
| LightGBM (T+E) | $1400_{-41,+41}$ | 0.275 | 24.0 | 18.9 | 0.0 | 13.4% | 416.56 | 2.24 |
| TabM (T+E) | $1399_{-43,+64}$ | 0.333 | 24.0 | 15.4 | 0.0 | 12.7% | 3133.91 | 1.27 |
| TabDPT (T) | $1388_{-57,+75}$ | 0.355 | 24.7 | 10.4 | 0.0 | 12.9% | 6154.73 | 39.45 |
| CatBoost (T+E) | $1388_{-45,+60}$ | 0.291 | 24.7 | 16.9 | 0.0 | 13.0% | 1665.53 | 0.56 |
| ModernNCA (T+E) | $1371_{-59,+76}$ | 0.343 | 25.9 | 13.1 | 1.0 | 13.5% | 4618.50 | 7.74 |
| CatBoost (T) | $1364_{-48,+47}$ | 0.265 | 26.4 | 17.6 | 0.0 | 13.4% | 1665.53 | 0.07 |
| XGBoost (T+E) | $1360_{-46,+43}$ | 0.231 | 26.7 | 19.2 | 0.0 | 14.0% | 700.96 | 1.44 |
| CatBoost (D) | $1343_{-47,+42}$ | 0.229 | 27.8 | 18.5 | 0.0 | 14.0% | 6.70 | 0.09 |
| LightGBM (T) | $1342_{-47,+50}$ | 0.221 | 27.9 | 22.9 | 0.0 | 14.3% | 416.56 | 0.38 |
| ModernNCA (T) | $1341_{-57,+62}$ | 0.263 | 28.0 | 14.7 | 1.0 | 14.1% | 4618.50 | 0.47 |
| TabM (T) | $1339_{-56,+62}$ | 0.261 | 28.1 | 17.9 | 0.0 | 13.6% | 3133.91 | 0.13 |
| XGBoost (T) | $1339_{-49,+46}$ | 0.207 | 28.1 | 20.0 | 0.0 | 14.1% | 700.96 | 0.21 |
| xRFM (T+E) | $1334_{-49,+60}$ | 0.252 | 28.5 | 18.0 | 0.0 | 14.1% | 866.11 | 2.01 |
| TabPFNv2 (T+E) | $1328_{-72,+75}$ | 0.312 | 28.9 | 11.4 | 1.0 | 14.9% | 2942.08 | 17.37 |
| Mitra (D) | $1301_{-70,+56}$ | 0.252 | 30.8 | 14.5 | 1.0 | 15.6% | 87.34 | 2.43 |
| xRFM (T) | $1290_{-45,+56}$ | 0.184 | 31.6 | 16.5 | 1.0 | 15.5% | 866.11 | 0.10 |
| TabDPT (D) | $1284_{-70,+72}$ | 0.253 | 32.0 | 16.4 | 0.0 | 15.4% | 45.42 | 39.41 |
| TabM (D) | $1278_{-51,+50}$ | 0.199 | 32.5 | 24.1 | 0.0 | 15.6% | 11.56 | 0.13 |
| TabICL (D) | $1278_{-60,+59}$ | 0.223 | 32.5 | 13.4 | 1.0 | 15.3% | 6.86 | 1.52 |
| EBM (T+E) | $1264_{-52,+50}$ | 0.156 | 33.4 | 24.3 | 0.0 | 16.8% | 2961.52 | 0.48 |
| TabPFNv2 (T) | $1260_{-62,+78}$ | 0.214 | 33.8 | 19.8 | 0.0 | 16.4% | 2942.08 | 0.26 |
| RealMLP (D) | $1253_{-50,+43}$ | 0.116 | 34.2 | 26.2 | 0.0 | 16.2% | 10.44 | *1.71* |
| TorchMLP (T+E) | $1253_{-54,+46}$ | 0.138 | 34.2 | 28.6 | 0.0 | 15.5% | 2832.80 | 1.80 |
| TabPFNv2 (D) | $1228_{-79,+66}$ | 0.189 | 36.0 | 18.6 | 0.0 | 16.9% | 3.27 | 0.32 |
| EBM (T) | $1218_{-57,+56}$ | 0.114 | 36.8 | 26.4 | 0.0 | 17.7% | 2961.52 | **0.05** |
| ModernNCA (D) | $1215_{-49,+59}$ | 0.114 | 37.0 | 18.5 | 1.0 | 18.3% | 13.74 | 0.32 |
| ExtraTrees (T+E) | $1198_{-60,+53}$ | 0.100 | 38.2 | 28.5 | 0.0 | 18.4% | 191.44 | 0.76 |
| EBM (D) | $1194_{-60,+57}$ | 0.111 | 38.5 | 20.1 | 1.0 | 18.5% | 7.66 | **0.05** |
| TorchMLP (T) | $1192_{-61,+48}$ | 0.102 | 38.6 | 30.9 | 0.0 | 17.3% | 2832.80 | 0.11 |
| XGBoost (D) | $1187_{-54,+55}$ | 0.098 | 39.0 | 20.6 | 1.0 | 17.4% | *2.06* | 0.12 |
| FastaiMLP (T+E) | $1173_{-71,+68}$ | 0.096 | 40.0 | 30.4 | 0.0 | 18.9% | 594.95 | 4.65 |
| ExtraTrees (T) | $1168_{-64,+66}$ | 0.101 | 40.3 | 26.5 | 0.0 | 19.4% | 191.44 | 0.10 |
| RandomForest (T+E) | $1163_{-64,+60}$ | 0.082 | 40.7 | 32.0 | 0.0 | 19.4% | 377.08 | 0.75 |
| LightGBM (D) | $1154_{-46,+48}$ | 0.071 | 41.2 | 36.2 | 0.0 | 18.1% | 2.20 | 0.17 |
| RandomForest (T) | $1121_{-46,+51}$ | 0.047 | 43.5 | 37.0 | 0.0 | 20.2% | 377.08 | 0.09 |
| FastaiMLP (T) | $1105_{-74,+62}$ | 0.060 | 44.5 | 32.5 | 0.0 | 20.5% | 594.95 | 0.34 |
| PerpetualBooster (T+E) | $1084_{-70,+60}$ | 0.066 | 45.9 | 37.6 | 0.0 | 25.4% | 176.26 | 0.50 |
| TabSTAR (T) | $1078_{-92,+82}$ | 0.109 | 46.3 | 18.7 | 0.0 | 24.4% | 39913.02 | 4.29 |
| TabSTAR (T+E) | $1075_{-99,+83}$ | 0.108 | 46.5 | 16.4 | 1.0 | 24.5% | 39913.02 | 20.17 |
| TorchMLP (D) | $1036_{-63,+54}$ | 0.023 | 48.8 | 44.9 | 0.0 | 22.2% | 8.96 | 0.13 |
| PerpetualBooster (T) | $1035_{-66,+55}$ | 0.035 | 48.9 | 44.6 | 0.0 | 26.8% | 176.26 | 0.19 |
| xRFM (D) | $1029_{-73,+73}$ | 0.049 | 49.3 | 39.2 | 0.0 | 25.0% | 3.14 | 0.74 |
| RandomForest (D) | $1000_{-58,+58}$ | 0.013 | 50.9 | 44.5 | 0.0 | 25.0% | *0.43* | **0.05** |
| TabSTAR (D) | $980_{-115,+106}$ | 0.081 | 52.0 | 19.5 | 1.0 | 30.1% | 398.77 | 4.65 |
| FastaiMLP (D) | $975_{-75,+66}$ | 0.019 | 52.2 | 48.8 | 0.0 | 24.3% | 3.12 | 0.31 |
| KNN (T+E) | $974_{-82,+63}$ | 0.024 | 52.3 | 46.5 | 0.0 | 26.6% | 129.10 | 1.63 |
| ExtraTrees (D) | $972_{-79,+71}$ | 0.014 | 52.4 | 48.5 | 0.0 | 26.4% | 0.26 | **0.05** |
| PerpetualBooster (D) | $938_{-91,+69}$ | 0.027 | 54.1 | 42.9 | 0.0 | 31.9% | 22.75 | 0.02 |

*(continues on next page)*

*(continued)*

| Model | Elo ($\uparrow$) | Norm. score ($\uparrow$) | Avg. rank ($\downarrow$) | Harm. mean rank ($\downarrow$) | #wins ($\uparrow$) | Improva-bility ($\downarrow$) | Train time per 1K [s] | Predict time per 1K [s] |
|---|---|---|---|---|---|---|---|---|
| Linear (T+E) | $901_{-98,+81}$ | 0.023 | 55.8 | 26.6 | 1.0 | 33.4% | 240.73 | 0.31 |
| Linear (T) | $875_{-114,+84}$ | 0.017 | 56.9 | 36.7 | 0.0 | 33.9% | 240.73 | 0.07 |
| KNN (T) | $824_{-92,+70}$ | 0.013 | 58.8 | 56.1 | 0.0 | 32.6% | 129.10 | 0.10 |
| Linear (D) | $815_{-106,+84}$ | 0.005 | 59.2 | 56.4 | 0.0 | 36.4% | 1.23 | 0.12 |
| KNN (D) | $619_{-109,+74}$ | 0.000 | 64.1 | 63.7 | 0.0 | 45.1% | **0.19** | **0.04** |

# K. Evaluation on the TALENT benchmark

We run *TabICLv2-Retouche* on the classification subset of the TALENT benchmark of Liu et al. (2024) restricted to datasets with at most 10 classes (170 binary and multiclass classification datasets). For this experiment **we use only the default configuration** for *TabICLv2-Retouche*.

**Setup**: We run three random training seeds per dataset and reuse the preprocessing pipeline of the TabArena-Lite headline run (Section 3). The main baseline is the unmodified TabICLv2 default, also at three seeds. The remaining baselines are the per-method, per-task numbers shipped with TALENT (Liu et al., 2024); we re-rank them in our pool rather than re-running them.

*TabICLv2-Retouche* completes all three seeds on every one of the 170 in-scope datasets. Ten of those datasets, in the upper tail of sample size or feature count, required a tighter context budget than the headline default in order to fit the differentiable forward pass through TabICLv2 within A100 memory and the CUDA kernel-grid limit. We cap the per-epoch in-context window at `max_samples = 15,000` to `60,000` rows depending on the dataset (chosen by halving from 60,000 until the forward fits), and on the two largest datasets we additionally chunk the query side at `max_query = 8,192` to keep `train+query` tokens below the 65,535 kernel-grid limit. The unmodified TabICLv2 baseline reaches comparable in-context budgets via its stock sklearn predict pipeline, which subsamples internally before each forward (Qu et al., 2026), so the deviation is approximately iso-context with the baseline rather than an asymmetric advantage.

**Result**: Table 4 reports the top mean per-task rank on Accuracy over the 170 in-scope datasets. *TabICLv2-Retouche* is the top-ranked method on the leaderboard at mean rank 3.83, ahead of the TabICLv2 base (4.05) and well clear of the next-best non-TabICL method TabR (9.32, 161 datasets); every other entry on the leaderboard sits at mean rank $\geq$ 9.8.

Head-to-head against the backbone base (*TabICLv2-Retouche* vs. TabICLv2): Table 5 reports per-metric paired wins, ties, and losses across the 170 datasets. *TabICLv2-Retouche* records more wins than losses on every metric. The aggregate over the 1,020 paired cells is 540 wins, 52 ties, and 428 losses. Restricted to the cells where the two methods produce different seed-averaged scores (i.e. ties are excluded), the per-metric win rate is 53.8% on Accuracy, 58.4% on AUC, 52.8% on Avg. Precision, 53.8% on Avg. Recall, 50.9% on F1, and 64.1% on LogLoss. The F1 margin is the smallest (81 wins vs. 78 losses, a 3-dataset edge). The gap on the remaining metrics is more substantial, with LogLoss the most decisive.

*Table 4.* Mean rank on the TALENT classification subset (Accuracy, 170 datasets with at most 10 classes). Methods shipped with the TALENT toolbox are re-ranked over this dataset universe; their own coverage may be a subset, in which case the column $n$ reflects that subset. Best per column in **bold**.

| Metric | Method | Mean rank ($\downarrow$) | n |
|---|---|---|---|
| Accuracy | ***TabICLv2-Retouche*** | 3.83 | 170 |
| Accuracy | TabICLv2 | 4.05 | 170 |
| Accuracy | TabR | 9.32 | 161 |
| Accuracy | RealMLP | 9.82 | 161 |
| Accuracy | ModernNCA | 10.10 | 159 |
| Accuracy | CatBoost | 10.42 | 161 |
| Accuracy | LightGBM | 10.78 | 161 |
| Accuracy | XGBoost | 11.97 | 161 |
| Accuracy | FTT | 12.88 | 161 |
| Accuracy | MLP_PRL | 13.30 | 161 |
| Accuracy | Resnet | 14.94 | 161 |

*Table 5.* Paired wins / ties / losses for *TabICLv2-Retouche* vs. the TabICLv2 default on the TALENT classification subset restricted to datasets with at most 10 classes (170 datasets per metric). "Win" means *TabICLv2-Retouche* strictly improves the seed-averaged metric over the base on that dataset.

| Metric | Wins | Ties | Losses |
|---|---|---|---|
| Accuracy | 84 | 14 | 72 |
| AUC | 97 | 4 | 69 |
| Avg. Precision | 84 | 11 | 75 |
| Avg. Recall | 85 | 12 | 73 |
| F1 | 81 | 11 | 78 |
| LogLoss | 109 | 0 | 61 |
| **Total** | **540** | **52** | **428** |

