# OpenReview forum: "TFM-Retouche: A Lightweight Input-Space Adapter for Tabular Foundation Models"
_ICML.cc/2026/Workshop/FMSD — FMSD @ ICML 2026 Poster_

### Official Review · Reviewer_GGB4 · 2026-05-13
**A Simple and Effective Input-Space Adaptation Strategy for Tabular Foundation Models**

**Rating:** 6
**Confidence:** 4

**Review:**

## Summary
This paper proposes TFM-Retouche, a lightweight input-space residual adapter for tabular foundation models. The method applies a gated near-identity transformation to the input while keeping the backbone frozen, together with an identity guard that falls back to the original model if adaptation is not beneficial.

## Strengths
- Simple and elegant adaptation strategy that avoids modifying pretrained backbone weights.
- The gated residual formulation is well motivated and practically appealing
- Strong empirical results on TabArena-Lite with thorough ablations and comparisons against LoRA, full fine-tuning, and BETA.
- The identity guard is a particularly nice practical contribution and appears genuinely important in the experiments.
- Good clarity overall, with solid implementation and experimental details.

## Weaknesses
- The main evaluation focuses on a single backbone (TabICLv2), which weakens the architecture-agnostic claim.
- Calibration-related metrics are not evaluated despite being part of the motivation.
- Some protocol details around validation reuse for HPO and guard decisions could be clarified further.
- The additional projection layer for high-dimensional datasets is not sufficiently isolated in the ablations.

## Overall Assessment
I found the paper technically solid and practically meaningful. The core idea is simple but effective, and the empirical results are strong given the modest adaptation complexity. The identity-guard mechanism is especially compelling from a deployment perspective. While broader backbone validation would strengthen the paper further, I believe this is a valuable contribution to tabular foundation model adaptation.

---

### Official Review · Reviewer_ivAh · 2026-05-19
**Promising Input Adapter for TFMs**

**Rating:** 6
**Confidence:** 5

**Review:**

# Summary
This paper proposes TFM-Retouche, an input-space adapter for tabular foundation models. Instead of fine-tuning the TFM backbone, it learns a small residual adapter of the input features and passes them into a frozen TabICLv2. The adapter is initialized near identity and uses a validation-based identity guard to fall back to the frozen model when adaptation does not help. On TabArena-Lite, the method reports the highest Elo, improving over frozen TabICLv2 by +56 Elo in the TE setting.

# Strengths
The paper addresses an important problem: how to adapt tabular foundation models without expensive fine-tuning. The input-space formulation is simple, practical, and architecture-agnostic in principle. It preserves the frozen TFM prior and avoids model-specific choices required by LoRA or other PEFT methods. The near-identity residual design is well motivated. The identity guard is also useful, since the adapter does not always help; the paper reports fallback in about one-third of dataset/config/fold runs.

# Areas for Improvement
1. The result should be framed more carefully. Although Retouche has the highest reported Elo, its confidence interval overlaps with other top methods, including TabPFN-2.6, RealTabPFN-2.5, and frozen TabICLv2.
2. The identity guard should be more prominently ablated. Since it falls back often (34% of the time), the main table should report Retouche with and without the guard.
3. The choice of adapter is plausible. A DCNv2-style cross block is reasonable, but the paper should compare against simpler input-space adapters, for example LoRA-like low-rank residual adapter.
4. The validation protocol should be clarified for the pure TFM inference mode. Specifically whether the validation is used as labeled context for the pure TFM models without fine-tuning.

# Detailed Comments
1. Please add a main-table comparison of Retouche with vs. without the identity guard.
2. Please include ablate with other simpler adapter baselines
3. Please clarify the context/query construction used when scoring the identity guard.
4. Please add a controlled ablation over DCN depth, e.g. L=0,1,2,3.
5. Please soften claims of state-of-the-art performance unless there is no longer an overlapping in the confidence intervals.

# Justification of Score
Score: 6/10 — weak accept

---

### Official Review · Reviewer_WXiT · 2026-05-20

**Rating:** 9
**Confidence:** 5

**Review:**

## Summary

This paper proposes TFM-Retouche, a lightweight method for adapting an input space before feeding it into a TFM, presumably aligning the incoming unseen data better with the inductive biases of the network. The method relies on a frozen base TFM, applying a learned, dimension-preserving transformation at the beginning of the network to produce better downstream predictions, allowing the pipeline to also "opt out" of this adaptation if it does not improve results on a holdout set.


## Strengths

I really liked reading this paper. It really piqued my interest and opened up a lot of questions below. This is a very good submission for the workshop. Some more notes:
- Very strong results on the TabArena-Lite baseline
- Good motivation, intuitive method, good exposition
- Timely technique with TFMs becoming increasingly popular


## Areas for Improvement

There are a lot of comments in the section below that could be considered improvement areas, although many could be considered minor (especially for a workshop).

One of the main things though is that I would like to see more information supporting the claim that the input space gets better aligned with the inductive biases of the network, beyond just showing better performance (which could presumably happen for other reasons than just what is claimed in the paper). I see there is something along these lines in the appendix, but it does leave me wanting more. I'm not sure what exactly would scratch this itch, though, admittedly.

## Detailed Comments
Not in any particular order

- Is this approach more applicable for models trained on synthetic data, like TabICL and TabPFN, than real data, like Real-TabPFN or TabDPT? It would be interesting to study the differences between the two regimes and how it relates to the alignment of unseen tasks.
- In the formulation, shouldn't we be comparing $f_\theta(g_\phi(X^\text{raw}))$ with $f_\theta(X)$, or at least the former with $f_\theta(X^\text{raw})$? It doesn't make sense to me that only $g_\phi$ would accept the pre-processed $X$.
- How truly lightweight is the adapter if we are required to do a full backward pass on the frozen TFM? Could this end up causing problems as TFMs get larger?
- Would it make more sense to match the objective of TFM-Retouche to the objective that the TFM was trained on for regression? For example, could you get better results using the pinball loss with TabICLv2 or the histogram loss with TabPFN 2+?
- Why is the TabDPT reference 2026? The model hit arXiv in Nov 2024 and was then published at NeurIPS 2025.
- How much hyper tuning is required for the fine-tuning module to work well? Was there a lot of effort put in to find decent "default" hypers before performing the full TabArena-Lite evaluation?
- There's a missing reference to perhaps the earliest fine-tuning work on TFMs: Retrieval & Fine-Tuning for In-Context Tabular Models, Thomas et al., NeurIPS, 2024 (https://arxiv.org/abs/2406.05207)
- It would be interesting to know how frequently the Identity Guard was deployed incorrectly, i.e., the validation set said don't use the adapted inputs, but it was actually the wrong choice for downstream performance.

## Justification

Very strong and interesting work. Somewhat limited in depth of contribution but still impactful. 100% belongs in this workshop and likely could become a full conference paper.